# Prognostic Value of PlGF Upregulation in Prostate Cancer

**DOI:** 10.3390/biomedicines12102194

**Published:** 2024-09-26

**Authors:** Manuel Scimeca, Erica Giacobbi, Francesca Servadei, Valeria Palumbo, Camilla Palumbo, Enrico Finazzi-Agrò, Simone Albisinni, Alessandro Mauriello, Loredana Albonici

**Affiliations:** 1Department of Experimental Medicine, TOR, University of Rome Tor Vergata, 00133 Rome, Italy; erica.giacobbi@gmail.com (E.G.); francescaservadei@gmail.com (F.S.); valeria.palumbo95@hotmail.com (V.P.); alessandro.mauriello@uniroma2.it (A.M.); 2Department of Clinical Sciences and Translational Medicine, University of Rome “Tor Vergata”, 00133 Rome, Italy; camilla.palumbo@uniroma2.it; 3Unit of Urology, Department of Surgical Sciences, Tor Vergata University, 00133 Rome, Italy; finazzi.agro@med.uniroma2.it (E.F.-A.); simone.albisinni@uniroma2.it (S.A.); 4Department of Biomedical Sciences, “Our Lady of Good Counsel” University, Rruga Dritan Hoxha, 1000 Tirana, Albania

**Keywords:** prostate cancer, bone metastasis, PlGF, VEGFR1, M2 macrophages, PD-1

## Abstract

Background: Prostate cancer (PCa) is the second most commonly diagnosed cancer in men worldwide, with metastasis, particularly to bone, being the primary cause of mortality. Currently, prognostic markers like PSA levels and Gleason classification are limited in predicting metastasis, emphasizing the need for novel clinical biomarkers. New molecules predicting tumor progression have been identified over time. Some, such as the immune checkpoint inhibitors (ICIs) PD-1/PD-L1, have become valid markers as theranostic tools essential for prognosis and drug target therapy. However, despite the success of ICIs as an anti-cancer therapy for solid tumors, their efficacy in treating bone metastases has mainly proven ineffective, suggesting intrinsic resistance to this therapy in the bone microenvironment. This study explores the potential of immunological intratumoral biomarkers, focusing on placental growth factor (PlGF), Vascular Endothelial Growth Factor Receptor 1 (VEGFR1), and Programmed Cell Death Protein 1 (PD-1), in predicting bone metastasis formation. Methods: we analyzed PCa samples from patients with and without metastasis by immunohistochemical analysis. Results: Results revealed that PlGF expression is significantly higher in primary tumors of patients that developed metastasis within five years from the histological diagnosis. Additionally, PlGF expression correlates with increased VEGFR1 and PD-1 levels, as well as the presence of intratumoral M2 macrophages. Conclusions: These findings suggest that PlGF contributes to an immunosuppressive environment, thus favoring tumor progression and metastatic process. Results here highlight the potential of integrating these molecular markers with existing prognostic tools to enhance the accuracy of metastasis prediction in PCa. By identifying patients at risk for metastasis, clinicians can tailor treatment strategies more effectively, potentially improving survival outcomes and quality of life. This study underscores the importance of further research into the role of intratumoral biomarkers in PCa management.

## 1. Introduction

Prostate cancer (PCa) is the second most diagnosed cancer in men worldwide, and metastatic disease is the principal cause of prostate-cancer-related mortality [1]. The most common site of metastasis from PCa is the bone [2,3]. Only 30% of men with metastatic PCa survive more than five years after diagnosis. Early-stage skeletal lesions are generally not detectable by current diagnostic tools. The sensitivity and specificity of these diagnostic tools are further limited when lesions progress slowly and may mimic non-malignant conditions. Consequently, the early detection of cancer cells’ molecular characteristics of localized disease could offer the identification of new theranostic tools useful in both prognosis and expanding therapeutic options, thereby improving the life expectancy of patients with PCa while preventing the onset of metastatic disease [3,4,5,6].

The mechanisms underlying the development of PCa metastasis involve crosstalk between tumor cells, the tumor microenvironment, and immune response cells [7,8,9]. The primary PCa cells actively modify the microenvironment of the site of future metastases before metastatic spread even occurs by inducing the formation of a supportive niche for circulating tumor cells and making it suitable for PCa cell implantation and growth [8,10]. However, bone metastasis (BM) detection often occurs late in tumor progression, making identifying early markers imperative.

It has been proposed that the following characteristics are required in the pre-metastatic niche to encourage the colonization of tumor cells and promote metastasis. Such factors include inflammation, angiogenesis/vascular permeability, lymphangiogenesis, immunosuppression, organotropism, reprogramming [11], and determining whether circulating tumor cells can colonize and survive in the new site. In this contest, several critical features of reactive tumor stroma, including stem/progenitor cells, inflammatory mediators, hypoxia, angiogenesis regulators, and exosomes, are accountable for promoting the creation of the metastatic niche [12,13,14]. Among the many factors involved in this intricate crosstalk, which could support tumor growth and progression, an important role could be played by the angiogenic factor, i.e., placental growth factor (PlGF). PlGF is a pleiotropic cytokine belonging to the vascular endothelial growth factor (VEGF) family, and its angiogenic/permeability activity is restricted to pathological conditions since PlGF expression is low to undetectable in most tissues in normal health [15]. PlGF signaling is mediated by binding VEGFR1 and the co-receptors neuropilin-1 (NRP1) and neuropilin-2 (NRP2), and the overexpression of these receptors in different tumor types contributes to angiogenesis, extracellular matrix (ECM) invasion, epithelial/mesenchymal transition, and resistance to anti-VEGF-A therapies [16,17,18,19]. PlGF promotes tumor progression by directly stimulating angiogenesis and tumor cell growth. In addition to tumor cells, many other cell types in the tumor microenvironment express PlGF, including endothelial cells, fibroblasts, tumor-associated macrophages (TAMs), and inflammatory cells. PlGF overexpression is induced by hypoxia, growth factors, and hormones [16].

PlGF also exerts indirect proangiogenic effects by acting on nonvascular cells, promoting monocyte survival and migration, myeloid progenitor recruitment, and macrophage M2 polarization. Moreover, primary-tumor-derived PlGF mobilizes and recruits VEGFR1+ myeloid-derived cells to pre-metastatic sites [10,20,21].

In addition to promoting tumor invasion, PlGF exerts immunosuppressive effects by influencing immune cell function. This contributes to the recruitment of tumor-associated macrophages (TAMs) [22] and myeloid-derived suppressor cells [23], reduces dendritic cell accumulation and impairs their maturation [24,25], and induces T-cell anergy and consequently cancer immune escape [26,27,28]. Finally, in a glioma tumor model, it has been reported that cancer-derived exosomes carrying PlGF, when captured by naive B cells, induced differentiation into Transforming Growth Factor β (TGF-β)+ Bregs that suppress the CD8+ T-cell activities [29].

Despite these effects, PlGF has not been closely associated with the progression of PCa. However, based on all the PlGF/VEGFR1 signaling features described above, we hypothesized that PlGF and VEGFR1 expression levels in PCa samples could potentially have a predictive value, allowing early identification of indolent PCa from potentially metastatic ones. Starting from these considerations, this study aimed to investigate the predictive value of PlGF expression and its associated signaling in identifying primary PCa with a high propensity to form bone metastases.

## 2. Materials and Methods

### 2.1. Sample Collection

A total of 53 prostate cancer samples were retrospectively collected from patients who underwent prostatectomy. The patients had an average age of 70.2 ± 1.1 years, ranging from 51 to 82 years. Residual paraffin blocs were used for histological classification and immunohistochemical investigations [30,31].

Specifically, serial sections from each sample were stained with hematoxylin and eosin for histological classification or incubated with the antibodies reported in Table 1 for immunohistochemical analyses.

The study was approved by the Institutional Ethical Committee of the “Policlinico Tor Vergata” (reference number # 120-23). All experimental procedures were conducted by the Code of Ethics of the World Medical Association, specifically the Declaration of Helsinki. 

### 2.2. Histology

Prostate tissues were paraffin-embedded after fixation in 10% buffered formalin for 24 h. Four-micrometer-thick sections were stained with hematoxylin and eosin (H&E).

### 2.3. Immunohistochemistry

Immunohistochemical analyses were conducted to study the possible role of PLGF in prostate cancer development by modulating the inflammatory response. Briefly, sections were subjected to antigen retrieval by treating them with EDTA citrate pH 7.8 at 95 °C for 30 min. Subsequently, the sections were incubated with the antibodies reported in Table 1 for 1 h at room temperature. Washing was performed using PBS/Tween20 pH 7.6. The reactions were visualized using the HRP-DAB Detection Kit (UCS Diagnostic, Rome, Italy). The immunoreaction was assessed by quantifying the number of positive cancer cells out of a total of 500 for PLGF, Hypoxia Inducible Factor 1alfa (HIF1a), and VGFR1. For the inflammatory infiltrate in the peritumoral areas, cluster of differentiation (CD)38, CD68, CD163, and Programmed Cell Death Protein 1 (PD-1) were used. Negative (no primary antibody incubation) and positive controls were utilized for each reaction. Immunohistochemical examination was independently performed on randomly selected areas of H&E sections by three pathologists (EG, FS, and AM) blinded to the clinical data. Interobserver reliability was greater than 98%.

### 2.4. Kaplan–Meier Plotter Analysis

To investigate the potential prognostic value of PIGF, the TNMplot tool of the Kaplan–Meier Plotter was employed to compare the expression of the genes PIGF, HIF1α, and FLT1 (VEGFR1 gene) in normal prostate, prostatic cancer, and metastasis [32]. The TNMplot graph refers to 106 normal samples, 283 prostatic cancer tissues, and six metastases. The gene signature function was used to study the concomitant expression of PIGF, HIF1α, and FLT1 genes in normal prostate, prostatic cancer, and metastasis. The Hallmark Enrichment Plot for PIGF, HIF1α, FLT1, and PD1 was obtained by uncovering cancer hallmarks tolls of TNMplot (core cancer hallmark gene set *n* = 1574) [33].

### 2.5. Statistical Analysis

The expression of investigated immunohistochemical biomarkers in the PCM+ and PCM- groups was studied using the Mann–Whitney U test. The correlation among biomarkers was analyzed using linear regression analysis. Kruskal–Wallis one-way variance analysis was applied to test whether samples originate from the same distribution in bioinformatics analysis.

## 3. Results

### 3.1. Histological Classification

According to the last WHO report, collected samples were classified by three pathologists (EG, FS, and AM) as follows: 38 prostate cancer lesions from patients who did not develop metastasis during the 5-year follow-up period (PCM- group) and 15 prostate cancer lesions from patients who developed metastasis (either in lymph nodes or bone) during the follow-up (PCM+).

Regarding the Gleason group (GG) classification, in the PCM- group, there were 68.42% (26 out of 38 lesions) with GG II, 10.53% (4 out of 38 lesions) with GG III, 15.79% (6 out of 38 lesions) with GG IV, and 5.2% (2 out of 38 lesions) with GG V. In the PCM+ group, the percentages were as follows: 20% (3 out of 15 lesions) with GG II, 33.33% (5 out of 15 lesions) with GS III, 33.33% (5 out of 15 lesions) with GG IV, and 13.33% (2 out of 15 lesions) with GG V. Despite observing a trend where higher GG correlates with the formation of metastasis, it is noteworthy that some patients developed metastatic lesions even with lower GG values at the time of diagnosis.

### 3.2. PlGF Expression

Immunohistochemical analysis was performed to evaluate the potential role of PlGF in prostate cancer progression. Additionally, the expression of PlGF was correlated with the expression of known molecules involved in PlGF activities, such as VEGFR1 and HIF1a, as well as with the presence of peritumoral macrophages (CD38, CD68, and CD163) and PD1-positive lymphocytes.

A significant increase in PlGF expression in PCM+ as compared to PCM- was observed (PCM− 96.9 ± 15.9; PCM+ 314.4 ± 21.9; *p* < 0.0001) (Figure 1A–C).

Compared to the GG, the primary prognostic histological biomarker for prostate cancer, PlGF expression showed a significant group effect (one-way ANOVA *p* = 0.009). T-test analysis showed a substantial increase in PLGF expression in lesions with GG IV and V compared to well-differentiated lesions (GG II) (Figure 1D).

Immunohistochemical analysis demonstrated that the number of VEFGR1-positive cells significantly increased in PCM+ concerning PCM− (PCM− 123.6 ± 15.86; PCM+ 309.9 ± 21.4; *p* < 0.0001) (Figure 2A–C). Specifically, PCM+ lesions frequently showed very intense positivity in the cytoplasm of prostate PCa cells (Figure 2C). Notably, linear regression analysis showed a positive significant association between PlGF and VEGFR1 expression by prostate cancer cells (*p* < 0.0001; r^2^ 0.68) (Figure 2D).

A similar trend was observed for the expression of HIF-1α (PCM− 142.7 ± 13.9; PCM+ 243.0 ± 21.8; *p* = 0.0009) (Figure 3A–C) and its correlation with PLGF (r^2^ 0.40; *p* < 0.0001) (Figure 3D). The immunohistochemical signal in PCM- lesions appears either absent or mild (Figure 3A). In contrast, in PCM+ lesions, the signal was intense, showing positivity in both peritumoral inflammatory cells and tumor cells (Figure 3B). Asterisks indicate significance levels: * *p* < 0.05, ** *p* < 0.01, *** *p* < 0.001.

### 3.3. PLGF Expression and Inflammatory Response

To evaluate the possible involvement of PlGF expression in the antitumoral inflammatory response, its expression was correlated with the presence of macrophages (both M1 and M2) and PD1-positive lymphocytes.

Concerning the total macrophages, linear regression analysis showed no association between PlGF expression and the number of CD68 (Figure 4A), CD38 (Figure 4B), or CD163 (Figure 4C) macrophages. 

Considering only the presence of macrophages in the enrolled prostate cancer lesions, a significant increase in M2 macrophages (CD163-positive cells) was observed in PCM+ as compared to PCM− (PCM− 102.7 ± 13.7; PCM+ 163.3 ± 21.2; *p* = 0.008) (Figure 4C). No differences were observed for total or M1 macrophages. 

Linear regression analysis revealed a significant positive association between the presence of intratumoral PD-1-positive lymphocytes and PLGF (r^2^ 0.4; *p* < 0.0001) (Figure 5A). However, a robust, significant increase in PD-1-positive intratumoral lymphocytes was observed in the PCM+ group regarding PCM− one (PCM− 41.5 ± 7.2; PCM+ 101.4 ± 10.4; *p* < 0.0001) (Figure 4D).

### 3.4. Bioinformatics Analysis

The bioinformatics study performed by the GO functional analysis tool provided by the TNM plotter confirmed the increased expression of PIGF in prostate cancers compared to metastasis (see Figure 5A) (*p* = 5.14 × 10^−2^) Similar data have been observed for HIF-1α expression (Figure 5A). No consistent differences were observed by investigating the expression of VEGFR1 (FLT1 gene) instead (Figure 5A). The concomitant expression of PLGF, HIF-1α, and VEGFR1 (FLT1 gene) was significantly higher in metastasis compared to non-metastatic prostate cancers (Figure 5B) (*p* = 1.71 × 10^−1^). The combined expression of PLGF, HIF-1α, VEGFR1 (FLT1 gene), and PD1 was associated with several cancer hallmarks, regardless of cancer type, including tissue invasion and metastasis, sustained angiogenesis, persistent proliferative signaling, immune evasion, and reprogramming of energy metabolism (Figure 5C).

## 4. Discussion

Despite significant advances in combination therapies with various immunotherapeutic approaches, identifying prognostic markers is increasingly necessary to direct appropriate therapeutic choices for solid tumors. It remains a considerable challenge for oncologists. In this context, immunological intratumoral biomarkers of immune response and immune dysfunction are potentially valuable areas of research for improving PCa prognosis.

PCa is the second leading cause of cancer death in men. Approximately 25–30% of cases are classified as the aggressive subtype (GG > II) and are prone to metastatic progression [4]. However, current clinical and histological prognostic parameters used in managing PCa are known to lack specificity. For example, factors such as aging and inflammation can influence Prostatic Specific Antigen (PSA) levels for early screening and recurrence detection [34]. Although the Gleason score and its refinement into the Gleason group classification offer a clearer framework for clinicians and assist in treatment decision-making [35], several studies have shown that PCa lesions with the same Gleason group can exhibit diverse biological behaviors and responses to treatment [36,37].

Several factors may contribute to these shortcomings. The Gleason score primarily evaluates the architectural pattern of tumor cells but does not capture the molecular or genetic changes that might influence metastatic behavior. Additionally, the heterogeneous nature of PCa means that even within a tumor classified with a low Gleason score, there might be subclonal populations with more aggressive characteristics.

This highlights the need for complementary diagnostic tools and biomarkers that can more accurately predict the metastatic potential of PCa. There is currently a lack of predictive markers to differentiate indolent PCa from potentially metastatic PCa at an early stage, despite significant progress in developing diagnostic markers for PCa [1,7,38].

Patients with localized PCa have a long-term survival rate. However, metastatic PCa remains incurable, and failure to respond to therapy results in advanced disease mortality [1]. This reflects the significant genetic heterogeneity of the tumor and the complex interaction of immune cells, stromal cells, and PCa cells in the tumor microenvironment [5,38]. PCa develops in a microenvironment where stromal and immune-infiltrating cells drive tumor growth and progression [6,39,40].

The primary stromal cell type in PCa is carcinoma-associated fibroblasts (CAFs). CAFs play a critical role in the tumor microenvironment (TME). They interact with prostate cancer cells and alter their metabolism and drug sensitivity [41]. In this regard, Zins et al. reported that fibroblast-derived PlGF mRNA levels increased after androgen deprivation therapy in prostate cancer, and increased PlGF levels had a direct dose-dependent proliferative effect on human PC-3 prostate cancer cells in vitro. Again, in xenograft tumor models, intratumoral administration of murine PlGF siRNA reduced stromal-derived PlGF expression, tumor burden, and the number of Ki-67-positive proliferating cells associated with reduced vessel density [15].

In addition, molecules produced by PCa cells and inflammatory cells in the tumor microenvironment, such as PlGF, can recruit inflammatory/myeloid-derived cells and create an immunosuppressive environment that drives prostate carcinogenesis and prevents the development of an efficient immune response against PCa cells. This leads to a state of unresolved immune response by preventing the activation and expansion of specific antitumor cells that inhibit rather than stimulate immunity [21,42,43,44,45].

PlGF is an angiogenic factor upregulated during inflammation and tumor development and progression [46,47] and is associated with immunosuppression in several tumor models [24,25,26]. Although normal prostate cells do not express PlGF, we observed a significant increase in PlGF expression in PCM+ compared to PCM− samples. Furthermore, PlGF expression increases in lesions with GG ≥ III compared to well-differentiated lesions (GG II). This finding correlates with many VEGFR1-positive PCa cells, suggesting that progressive expression of PlGF by PCa cells and immune-infiltrating cells could be responsible for tumor promotion and progression.

In addition, previous research has reported that PlGF upregulates the transcription factor NFAT-1 and that the VEGFR1 promoter contains a binding site for NFAT-1, thus creating a circuit that amplifies the effect of PlGF [48]. Notably, NFAT-1 is involved in tumor cell survival, invasive migration, and tumor-induced CD4+ T-cell anergy by releasing pro-inflammatory cytokines, including PlGF [49].

Increased expression of PlGF and its receptor VEGFR1 is stimulated by the progressive hypoxia in the tumor core. Hypoxia also regulates macrophage recruitment through PlGF release [50], and macrophages migrate and accumulate in the most hypoxic regions of tumor tissue [46]. Moreover, hypoxia plays an essential role in the aggressiveness of PCa. Accordingly, the bioinformatics analysis showed that the concomitant expression of PlGF, VEGFR1, and HIF1a is significantly increased in metastatic PCas compared to non-metastatic ones. In this regard, Bharti et al. have elegantly documented the presence of hypoxia in primary and metastatic PCa sites and the crucial role of hypoxia in metastasis [51]. Indeed, PCa ranks high among malignancies in which hypoxia is critical for treatment resistance and metastasis [52]. Angiogenic growth factors and hypoxia have also been linked to tumor stem cell self-renewal and metastasis and have been identified as factors responsible for tumor-associated immunosuppression [53,54,55]. Inhibition of myeloid chemotaxis has been shown to reduce tumor-induced myeloid inflammation and reverse therapy resistance in a subset of patients with metastatic castration-resistant prostate cancer (CRPC) [56].

PlGF also promotes M2 macrophage polarization in TME [46]. These cells have been shown to play a critical role in different stages of PCa, accounting for up to 50% of the tumor mass [20]. TAM density was reported to be higher in metastatic than non-metastatic PCa cores, and increased TAM density at biopsy may predict a worse prognosis [20,57,58]. Thus, we evaluated PlGF expression in the intratumoral inflammatory infiltrate. Our results coherently show that PlGF expression in PCa specimens correlates with a significant increase in M2 macrophages (CD163+ cells) in PCM+ compared to PCM- samples. However, no difference was observed in the total number of macrophages. This result suggests that TAMs are functionally transformed from M1 to M2 by the PlGF-rich tumor microenvironment.

Hypoxia strongly also increases macrophage-mediated T-cell suppression in fashion-dependent HIF-1α macrophage expression [59]. The occurrence of different subsets of T cells in the tumor microenvironment depends on the patterns of cytokines and receptor expression. PlGF/VEGFR-1 signaling is also involved in the modulation of T cells’ immune response. Leplina et al. revealed that PlGF can modulate the human T-cell functions in a wide dose range, suppress peripheral blood mononuclear cell (PBMC) proliferation, and inhibit CD4^+^ and CD8^+^ T cells. VEGFR-1 blocking, but not VEGFR-2 with neutralizing Abs, completely abolished the suppressive effect of PlGF. Moreover, the PBMC treatment with PlGF upregulated IL-10 production in CD4^+^ and CD8^+^ T cells, promoted CD8^+^ T-cell apoptosis, and enhanced the expression of PD-1 and T-cell immunoglobulin and mucin-domain containing-3 (TIM-3) on activated T cells [60]. 

Hypoxia triggers rapid and robust upregulation of PD-L1 on MDSCs, macrophages, dendritic cells, and tumor cells in tumor-bearing mice, and this upregulation is specifically dependent on hypoxia-inducible factor 1 alpha (HIF-1 alpha) [61,62]. Hypoxia also upregulates the expression of programmed PD-L1 in PCa tissues compared to paired normal tissues, and elevated PD-L1 expression is associated with poor outcomes in PCa patients [63]. Moreover, high levels of PD-L1 expression are positively related to proliferation, the Gleason group, and androgen receptor expression in patients with aggressive primary PCa [64,65]. These reports led us to investigate PD-1 immune checkpoint molecule expression levels in PCa samples. We found a significant positive association between the presence of peritumoral PD1-positive lymphocytes and PlGF in the PCM+ group concerning PCM−. In addition, we found a positive association between the presence of intratumoral PD1-positive lymphocytes and PlGF expression level.

Finally, the simultaneous expression of HIF-1, PlGF, VEGFR1, and PD-1 in PCa determines the main features necessary for the establishment of the pre-metastatic niche, such as inflammation, immunosuppression, vascular angiogenesis/permeability, and reprogramming capable of inducing the formation of a supportive microenvironment that allows the colonization of tumor cells in a secondary organ.

## 5. Conclusions

The management of PCa remains a significant challenge for clinicians, mainly when accurately predicting disease progression and tailoring treatment strategies [66,67]. The current morphological and biochemical markers used to establish the prognosis of PCa frequently fail to predict metastasis formation. In this scenario, our study emphasizes the potential of reliable biomarkers as valuable tools for enhancing the accuracy of PCa prognosis and informing treatment strategies. Specifically, by integrating Gleason scoring with other molecular characteristics of PCa, such as the expression of PlGF, VEGFR1, and PD-1, clinicians can better tailor treatment plans and improve outcomes for patients at risk of developing metastases. Our findings suggest that PlGF, in particular, plays a critical role in promoting metastasis, likely through its involvement in angiogenesis, immune suppression, and the creation of a pre-metastatic niche. Elevated PlGF levels in primary tumors may be indicative of an aggressive phenotype with a higher propensity to form PCa metastasis. The study’s results highlight that monitoring PlGF expression, in conjunction with traditional markers, could provide more precise prognostic information, helping to identify patients with an elevated risk of metastasis early in their disease course. Therefore, incorporating PlGF expression as a prognostic tool could significantly improve the ability to predict metastatic potential and guide clinical decision-making for high-risk patients.

## Figures and Tables

**Figure 1 biomedicines-12-02194-f001:**
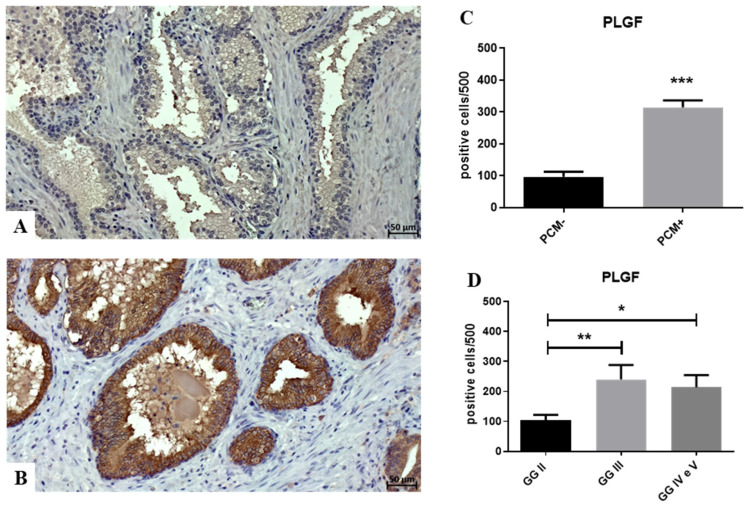
PlGF expression in prostate cancers. (**A**) No or rare PlGF-positive cancer cells are observed in PCM- lesions. (**B**) Numerous PlGF-positive cancer cells are present in a PCM+ sample. (**C**) The graph shows a significant increase in PlGF-positive cancer cells in the PCM+ group compared to the PCM− group. (**D**) The graph displays a significant distribution of PlGF-positive cancer cells among different Gleason groups (one-way ANOVA, *p* = 0.009). PlGF-positive cancer cells are significantly lower in GGII compared to other Gleason groups. Asterisks indicate significance levels: * *p* < 0.05, ** *p* < 0.01, *** *p* < 0.001.

**Figure 2 biomedicines-12-02194-f002:**
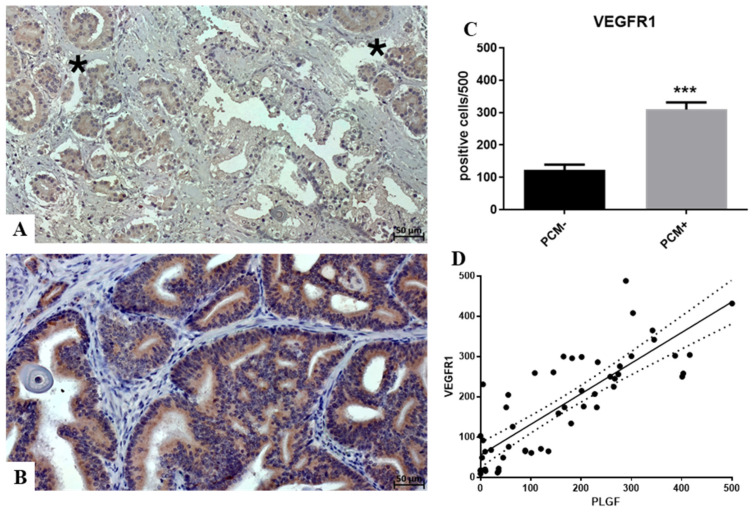
VEGFR1 expression and its association with PlGF. (**A**) Some faint VEGFR1-positive cancer cells are observed in PCM− lesions (asterisks). (**B**) Numerous VEGFR1-positive cancer cells are present in a PCM+ sample. (**C**) The graph shows a significant increase in VEGFR1-positive cancer cells in the PCM+ group compared to the PCM− group. (**D**) Linear regression analysis shows a significant positive association between the number of VEGFR1-positive and PlGF-positive prostate cancer cells. Asterisks indicate significance levels: *** *p* < 0.001.

**Figure 3 biomedicines-12-02194-f003:**
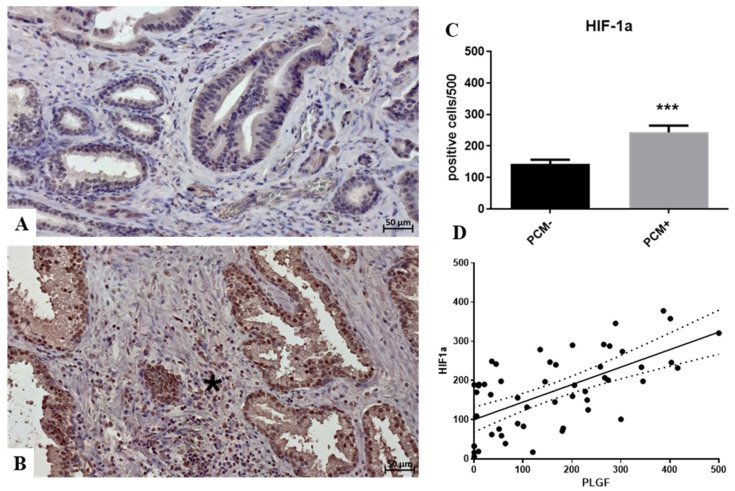
HIF1α expression and its association with PlGF. (**A**) Some faint HIF1α-positive cancer cells are observed in PCM− lesions. (**B**) Numerous PlGF-positive cancer cells are present in a PCM+ sample. The HIF1α reaction also stains inflammatory cells (asterisk). (**C**) The graph shows a significant increase in VEGFR1-positive cancer cells in the PCM+ group compared to the PCM− group. (**D**) Linear regression analysis shows a significant positive association between the number of HIF1α-positive and PlGF-positive prostate cancer cells. Asterisks indicate significance levels: *** *p* < 0.001.

**Figure 4 biomedicines-12-02194-f004:**
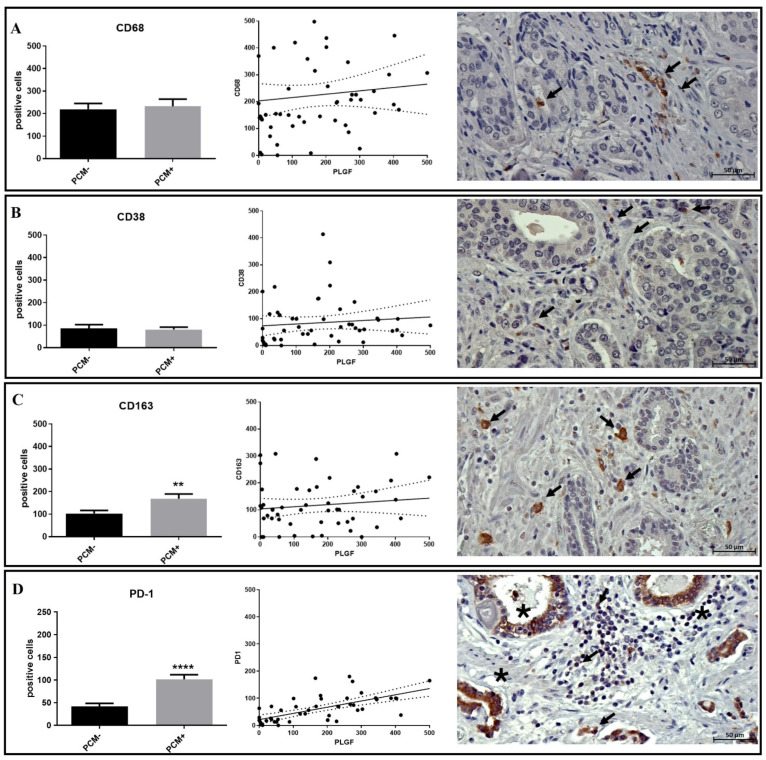
Investigation of immune cells and their association with PlGF expression. (**A**) CD68-positive macrophages do not increase in PCM+ lesions, nor are they associated with PlGF expression. The image shows CD68-positive cells in the peritumoral area of a PCM+ lesion (arrows). (**B**) CD38-positive macrophages do not increase in PCM+ lesions, nor are they associated with PlGF expression. The image shows CD38-positive macrophages in the peritumoral area of a PCM+ lesion (arrows). (**C**) CD163-positive macrophages significantly increase in PCM+ lesions compared to PCM− ones. The number of CD163-positive macrophages is not associated with PlGF expression. The image shows CD163-positive macrophages in the peritumoral area of a PCM+ lesion (arrows). (**D**) PD1-positive lymphocytes significantly increase in PCM+ lesions compared to PCM- ones. The number of PD1-positive lymphocytes is associated with PlGF expression. The image shows PD1-positive lymphocytes in the peritumoral area of a PCM+ lesion (arrows). PD1-positive cancer cells are highlighted (asterisks). Asterisks indicate significance levels: ** *p* < 0.01, **** *p* < 0.0001.

**Figure 5 biomedicines-12-02194-f005:**
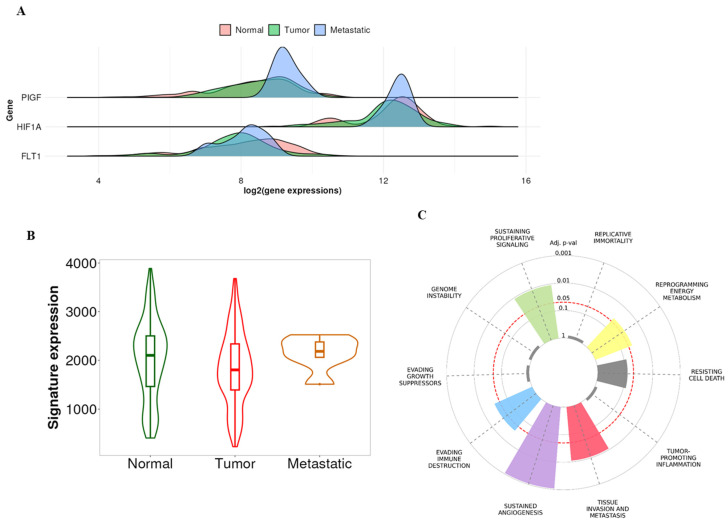
Bioinformatics analysis. (**A**) The graph shows a significant increase in both PlGF and HIF1α gene expression in metastasis compared to prostate cancer. Slight differences are observed for VEGFR1 (FLT1 gene). (**B**) The graph shows an increased expression of PlGF, HIF-1α, and VEGFR1 (FLT1 gene) in metastasis compared to prostate cancer. (**C**) The graph displays the potential role of PlGF, HIF-1α, VEGFR1 (FLT1 gene), and PD1 in cancer hallmarks.

**Table 1 biomedicines-12-02194-t001:** List of antibodies.

Antibody	Clone and Company	Dilution
PlGF	Rabbit Polyclonal, ab196666, Abcam, Cambridge, UK	1:100
VEGFR1	Mouse monoclonal; D2 clone; Santa Cruz Biotechnology, Dallas, TX, USA	1:250
HIF1α	Rabbit Monoclonal, clone EP1215Y; Abcam, Cambridge, UK	1:100
CD38	Rabbit monoclonal clone SP149; Ventana, Tucson, AZ, USA	Pre-diluted
CD68	Mouse monoclonal clone KP-1; Ventana, Tucson, AZ, USA	Pre-diluted
CD163	Mouse monoclonal clone MRQ-26; Ventana, Tucson, AZ, USA	Pre-diluted
PD1	Mouse monoclonal clone NAT105; Ventana, Tucson, AZ, USA	Pre-diluted

## Data Availability

The data will be made available upon reasonable request.

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
