# Peer review of "Prognostic Value of PlGF Upregulation in Prostate Cancer"

_biomedicines, 2024, doi:10.3390/biomedicines12102194_

Round 1
Reviewer 1 Report
Comments and Suggestions for Authors
The manuscript entitled “Prognostic Value of Plgf Upregulation in Prostate Cancer” was easy to follow. I have following comments/suggestions
1. Please perform the scoring of slides with the help of a pathologist and use the appropriate scores to correlate and then segregate based on the relevant features including the PCM grouping or Gleason’s scores
2. There would be a possibility of bias with just the cell count even though the fields were selected randomly
3. It is interesting to see that PLGF was not closely associated with progression of prostate cancer but the positive cell population correlate with HIF1alpha and VEGF1
4. Could you possibly use some datasets like TCGA to add the correlation analysis of PLGF, VEGF1 and HIF1alpha along with other other immune cell markers used in the study. This can provide evidence at RNA level and bolster your hypothesis.
Author Response
Manuscript ID biomedicines-3193945
" Prognostic Value of PlGF Upregulation in Prostate Cancer"
Submitted to: Biomedicines
Section: Molecular and Translational Medicine
Special issue: Angiogenic Growth Factors in Tumor Development: Beyond New
Blood Vessels Formation
POIT-TO-POINT REBUTTAL TO REVIEWERS COMMENTS
General Comments to Editor and Reviewers
We appreciated the thoughtful and constructive criticisms and suggestions of both Editor and Reviewers. His/her comments on how to improve the manuscript, which has been revised accordingly. We also appreciate the Editors for calling for a new re-submission of an improved version of our manuscript.
In the revised version, we have addressed the reviewers’ concerns, which resulted in a significantly strengthened manuscript. We have also added new bioinformatic results that further substantiate the relevance of our manuscript.
REVIEWER#1
The manuscript entitled “Prognostic Value of Plgf Upregulation in Prostate Cancer” was easy to follow. I have following comments/suggestions
Reply: Thank you for your thoughtful feedback and for appreciating our work. We value your suggestions and will carefully consider them to improve the manuscript.
- Please perform the scoring of slides with the help of a pathologist and use the appropriate scores to correlate and then segregate based on the relevant features including the PCM grouping or Gleason’s scores
- Reply: thanks for this point out. The valuation of both H&E and IHC sections were performed by pathologists (Erica Giacobbi, Francesca Servadei and Alessandro Mauriello). We specified this in the new version of the manuscript.
There would be a possibility of bias with just the cell count even though the fields were selected randomly.
Reply: The areas have been randomly selected by an expert pathologist on H&E sections. This procedure ensured the accurate observation of tumor areas without the influence of IHC signaling. We better explain this in the new version of the manuscript.
Methods section pag.3
Immunohistochemical examination was independently performed on randomly se-lected areas of H&E sections by three different pathologists (EG, FS, and AM), who were blinded to the clinical data. Interobserver reliability was greater than 98%.
- It is interesting to see that PLGF was not closely associated with progression of prostate cancer but the positive cell population correlate with HIF1alpha and VEGF1
- Could you possibly use some datasets like TCGA to add the correlation analysis of PLGF, VEGF1 and HIF1alpha along with other other immune cell markers used in the study. This can provide evidence at RNA level and bolster your hypothesis.
Reply points #3 and #4: we performed bioinformatic analysis through the GO functional analysis tool provided by the TNM plotter. PlGF expression by RNASeq in prostate cancers confirm the significant increase in its expression in metastatic cancers as compared to non-metastatic lesions (see figure 5). In addition, molecular data showed a great increase in HIF1a in metastatic lesions as compared to non-metastatic ones. No consistent differences were observed for VEGFR1 (FLT1 gene). The concomitants expression of molecules associated to PlgF in our study was associated to several cancer hallmarks including tissue invasion and metastasis, sustained angiogenesis, sustained proliferative signaling, evading immune destruction and reprogramming energy metabolisms.
The text was modified as follow:
Methods Pag.3
2.4 Bioinformatic analysis
To investigate the potential prognostic value of PIGF, the TNMplot tool of the Kaplan–Meier Plotter was employed to compare the expression of the genes PIGF, HIF1α, and FLT1 (VEGFR1 gene) in normal prostate, prostatic cancer, and metastasis [32]. TNMplot graph refers to 106 normal samples, 283 prostatic cancer tissues, and six metastases. The gene signature function was used to study the concomitant expression of PIGF, HIF1α, and FLT1 genes in normal prostate, prostatic cancer, and metastasis. Hallmark Enrich-ment Plot for PIGF, HIF1α, FLT1, and PD1 has been obtained by uncovering cancer hallmarks tolls of TNMplot (core cancer hallmark gene set n=1574) [33].
Results Pag.9
2.4 Bioinformatic analysis
Bioinformatic study performed by the GO functional analysis tool provided by the TNM plotter confirmed the increased expression of PIGF in prostate cancers as compared to metastasis (see figure 5A) (p=5.14e-02.) Similar data have been observed for HIF-1α expression (Figure 5A). No consistent differences were observed by investigating the expression of VEGFR1 (FLT1 gene) in-stead (Figure 5A). The concomitant expression of PLGF, HIF-1α, and VEGFR1 (FLT1 gene) was significantly higher in metastasis compared to non-metastatic prostate cancers (Figure. 5B) (p=1.71e-01). The combined expression of PLGF, HIF-1α, VEGFR1 (FLT1 gene), and PD1 was as-sociated with several cancer hallmarks, regardless of cancer type, including tissue invasion and metastasis, sustained angiogenesis, persistent proliferative signaling, immune evasion, and re-programming of energy metabolism (Figure. 5C).
Discussion Pag.15
Accordingly, bioinformatic analysis showed that the concomitant expression of PlGF, VEGFR1, and HIF1a is significantly increased in metastatic PCas compared to non-metastatic ones.
Bioinformatic data have been showed in the figure 5.
Figure 5. Bioinformatic analysis. A) The graph shows a significant increase in both PlGF and HIF1α gene expression in metastasis compared to prostate cancer. Slight differences are observed for VEGFR1 (FLT1 gene). B) The graph shows an increased expression of PlGF, HIF-1α, and VEGFR1 (FLT1 gene) in metastasis compared to prostate cancer. C) The graph displays the potential role of PlGF, HIF-1α, VEGFR1 (FLT1 gene), and PD1 in cancer hallmark

Reviewer 2 Report
Comments and Suggestions for Authors
This manuscript investigated the relationship between placental growth factor (PLGF) expression and prostate cancer (PCa) metastasis. The results showed that PIGF contributes to the immunosuppressive environment, thus favoring tumor progression and metastatic process. This finding is of great help to improve the accuracy of prostate cancer metastasis prediction and has important practical significance. However, the following issues should be addressed before the paper is considered suitable for publication in Biomedicines.
1. Full names should be given for abbreviations that appear for the first time in the article, for example: PSA, VEGFR1, PD-1, etc. Please modify this.
2. A total of 53 prostate cancer samples were collected in the article, and the sample size was not large enough. In addition, 38 cases of patients did not develop metastasis within 5 years, and 15 cases of prostate cancer patients metastasized, this ratio is unreasonable. It is recommended to increase the number of prostate cancer samples with metastasis.
3. The layout of figure 1, figure 2 and figure 3 is out of alignment, please re-typeset.
4. In the text, the relevant expressions of figure 2 and figure 3 are less, and the content is very empty. On the contrary, the figure captions are too lengthy, and lack a sense of proportion. It is recommended to put some of the content of the figure captions be placed in the body of the article.
5. In line 96, the format of “μm” is incorrect and should be changed to “μm”. Please check if there are similar problems in the text.
6. The topic of this article is the prognostic value of PLGF upregulation in prostate cancer, but the specific correlation between PIGF and prostate cancer metastasis was not be clearly pointed out in the conclusion of this article. Please elaborate on this.
7. The topic of the article is related to cancer, and it is suggested to introduce the latest cancer diagnosis and treatment methods in the article. Here are many recent articles for reference: Chem. Soc. Rev. 2021, 50, 2839; Exploration 2021, 1, 21; Asian J Pharm Sci. 2022, 17, 253; Molecules 2023, 28, 1470; Sci. China: Chem. 2023, 66, 613; Adv. Mater. 2024, 36, 2304249.
Comments on the Quality of English LanguageExtensive editing of English language required.
Author Response
Manuscript ID biomedicines-3193945
" Prognostic Value of PlGF Upregulation in Prostate Cancer"
Submitted to: Biomedicines
Section: Molecular and Translational Medicine
Special issue: Angiogenic Growth Factors in Tumor Development: Beyond New
Blood Vessels Formation
POIT-TO-POINT REBUTTAL TO REVIEWERS COMMENTS
General Comments to Editor and Reviewers
We appreciated the thoughtful and constructive criticisms and suggestions of both Editor and Reviewers. His/her comments on how to improve the manuscript, which has been revised accordingly. We also appreciate the Editors for calling for a new re-submission of an improved version of our manuscript.
In the revised version, we have addressed the reviewers’ concerns, which resulted in a significantly strengthened manuscript. We have also added new bioinformatic results that further substantiate the relevance of our manuscript.
REVIEWER#2
This manuscript investigated the relationship between placental growth factor (PLGF) expression and prostate cancer (PCa) metastasis. The results showed that PIGF contributes to the immunosuppressive environment, thus favoring tumor progression and metastatic process. This finding is of great help to improve the accuracy of prostate cancer metastasis prediction and has important practical significance. However, the following issues should be addressed before the paper is considered suitable for publication in Biomedicines.
Reply: Thank you for your thoughtful feedback and for appreciating our work. We value your suggestions and will carefully consider them to improve the manuscript.
- Full names should be given for abbreviations that appear for the first time in the article, for example: PSA, VEGFR1, PD-1, etc. Please modify this.
Reply: Thanks for this point out. We modified the manuscript according to reviewer suggestion.
- A total of 53 prostate cancer samples were collected in the article, and the sample size was not large enough. In addition, 38 cases of patients did not develop metastasis within 5 years, and 15 cases of prostate cancer patients metastasized, this ratio is unreasonable. It is recommended to increase the number of prostate cancer samples with metastasis.
Reply: We believe our 5-year follow-up offers reliable case series. The data collected during this period are enough to provide significant insights, allowing to identify possible biomarkers for prostate cancer such as PiGF as showed in the manuscript.
In terms of metastasis distribution at the 5-year follow-up, our results are consistent with what we typically observe at Policlinico Tor Vergata and in the literature (PMID: 36581423). In fact, a different proportion of metastatic and non-metastatic patients is generally observed at the 10-year follow-up, where a significant increase in metastasis formation has been noted. Data from the 5-year follow-up are important because, at this point, we can study more aggressive tumors capable of developing metastatic lesions early, within the first few years after diagnosis.
- The layout of figure 1, figure 2 and figure 3 is out of alignment, please re-typeset.
Reply: thanks for this point out. The figures 1, 2 and 3 have been properly re-typeset to correct the alignment issues.
- In the text, the relevant expressions of figure 2 and figure 3 are less, and the content is very empty. On the contrary, the figure captions are too lengthy, and lack a sense of proportion. It is recommended to put some of the content of the figure captions be placed in the body of the article.
Reply: Thanks for this suggestion. In the new version of our manuscript, we extended the description of immunohistochemical reactions for both VEGFR1 and HIF1a.
- In line 96, the format of “μm” is incorrect and should be changed to “μm”. Please check if there are similar problems in the text.
Reply: done
- The topic of this article is the prognostic value of PLGF upregulation in prostate cancer, but the specific correlation between PIGF and prostate cancer metastasis was not be clearly pointed out in the conclusion of this article. Please elaborate on this.
Reply: We agree with the reviewer about the possibility to improve the conclusions with further consideration about the prognostic value of PlGF in prostate cancer.
According to this, the conclusions have been extended as follow:
Conclusions pages 11 and 12
The management of PCa remains a significant challenge for clinicians, mainly when accurately predicting disease progression and tailoring treatment strategies [66,67]. The current morphological and biochemical markers used to establish the prognosis of PCa frequently fail to predict metastasis formation. In this scenario, our study emphasizes the potential of reliable biomarkers as valuable tools for enhancing the accuracy of PCa prognosis and informing treatment strategies. Specifically, by integrating Gleason scoring with other molecular characteristics of PCa, such as the expression of PlGF, VEGFR1, and PD-1, clinicians can better tailor treatment plans and improve outcomes for patients at risk of developing metastases. Our findings suggest that PlGF, in particular, plays a critical role in promoting metastasis likely through its involvement in angiogenesis, immune suppression, and the creation of a pre-metastatic niche. Elevated PlGF levels in primary tumors may be indicative of an aggressive phenotype with a higher propensity to form PCa metastasis. The study's results highlight that monitoring PlGF expression, in conjunction with traditional markers, could provide more precise prognostic information, helping to identify patients with an elevated risk of metastasis early in their disease course. Therefore, incorporating PlGF expression as a prognostic tool could significantly improve the ability to predict metastatic potential and guide clinical decision-making for high-risk patients.
- The topic of the article is related to cancer, and it is suggested to introduce the latest cancer diagnosis and treatment methods in the article. Here are many recent articles for reference: Chem. Soc. Rev. 2021, 50, 2839; Exploration 2021, 1, 21; Asian J Pharm Sci. 2022, 17, 253; Molecules 2023, 28, 1470; Sci. China: Chem. 2023, 66, 613; Adv. Mater. 2024, 36, 2304249.
Reply: "We noted that only some of the references are relevant to the aim of our manuscript, and we have cited two of them in the conclusion section. References #66 and #67.

Reviewer 3 Report
Comments and Suggestions for Authors
The manuscript “Prognostic Value of Plgf Upregulation in Prostate Cancer” was submitted by authors. I appreciated the authors to carry out such as nice work. The following suggestions may be consider for improving the manuscript. There few important corrections to be rectified.
1. Expand VEGFR1 and PD-1 in abstract.
2. There is continuity missing in introduction. Improve the introduction part.
3. Highlight the findings and future prospectives of present research work.
4. Elaborate about the other diagnostic and theranostic tools used to diagnose and cure prostate cancer in introduction part.
5. Kindly Change the funding details to “no funding” instead of “no founding”.
6. Research work was perfectly explained in a reader friendly language.
7. Images were of high quality and were brilliantly explained.
8. Overall a great research paper.
Author Response
Manuscript ID biomedicines-3193945
" Prognostic Value of PlGF Upregulation in Prostate Cancer"
Submitted to: Biomedicines
Section: Molecular and Translational Medicine
Special issue: Angiogenic Growth Factors in Tumor Development: Beyond New
Blood Vessels Formation
POIT-TO-POINT REBUTTAL TO REVIEWERS COMMENTS
General Comments to Editor and Reviewers
We appreciated the thoughtful and constructive criticisms and suggestions of both Editor and Reviewers. His/her comments on how to improve the manuscript, which has been revised accordingly. We also appreciate the Editors for calling for a new re-submission of an improved version of our manuscript.
In the revised version, we have addressed the reviewers’ concerns, which resulted in a significantly strengthened manuscript. We have also added new bioinformatic results that further substantiate the relevance of our manuscript.
REVIEWER#3
The manuscript “Prognostic Value of Plgf Upregulation in Prostate Cancer” was submitted by authors. I appreciated the authors to carry out such as nice work. The following suggestions may be consider for improving the manuscript. There few important corrections to be rectified.
- Expand VEGFR1 and PD-1 in abstract.
Reply: Thank you for your thoughtful feedback and for appreciating our work. We value your suggestions and will carefully consider them to improve the manuscript.
- There is continuity missing in introduction. Improve the introduction part.
Reply: Thanks for this point out. In the revised version of our manuscript the introduction has been improved by adding several molecular details.
The text was modified as follow:
Pag. 1
PlGF signaling is mediated by binding VEGFR1 and the co-receptors neuropilin-1 (NRP1) and neuropilin-2 (NRP2), and the overexpression of these receptors in different tumor types contributes to angiogenesis, extracellular matrix (ECM) invasion, epitheli-al/mesenchymal transition, and resistance to anti-VEGF-A therapies [16-19]. PlGF pro-motes tumor progression by directly stimulating angiogenesis and tumor cell growth. In addition to tumor cells, many other cell types in the tumor microenvironment express PlGF, including endothelial cells, fibroblasts, tumor-associated macrophages (TAMs), and inflammatory cells. PlGF overexpression is induced by hypoxia, growth factors, and hormones [16].
PlGF also exerts indirect proangiogenic effects by acting on nonvascular cells, promoting monocyte survival and migration, myeloid progenitor recruitment, and macrophage M2 polarization. Moreover, primary tumor-derived PlGF mobilizes and recruits VEGFR1+ myeloid-derived cells to pre-metastatic sites [10,20,21].
In addition to promoting tumor invasion, PlGF exerts immunosuppressive effects by influencing immune cell function. This contributes to the recruitment of tu-mor-associated macrophages (TAMs) [22] and myeloid-derived suppressor cells [23], reduces dendritic cell accumulation and impairs their maturation [24,25], induces T-cell anergy, and, consequently, cancer immune escape. [26-28].
- Highlight the findings and future prospectives of present research work.
Reply: thanks for this suggestion. In the new version of our manuscript, we emphasized the possible use of PlGF as prognostic biomarkers in the management of prostate cancer in the conclusions section.
Conclusions pages 11 and 12
The management of PCa remains a significant challenge for clinicians, mainly when accurately predicting disease progression and tailoring treatment strategies [66,67]. The current morphological and biochemical markers used to establish the prognosis of PCa frequently fail to predict metastasis formation. In this scenario, our study emphasizes the potential of reliable biomarkers as valuable tools for enhancing the accuracy of PCa prognosis and informing treatment strategies. Specifically, by integrating Gleason scoring with other molecular characteristics of PCa, such as the expression of PlGF, VEGFR1, and PD-1, clinicians can better tailor treatment plans and improve outcomes for patients at risk of developing metastases. Our findings suggest that PlGF, in particular, plays a critical role in promoting metastasis likely through its involvement in angiogenesis, immune suppression, and the creation of a pre-metastatic niche. Elevated PlGF levels in primary tumors may be indicative of an aggressive phenotype with a higher propensity to form PCa metastasis. The study's results highlight that monitoring PlGF expression, in conjunction with traditional markers, could provide more precise prognostic information, helping to identify patients with an elevated risk of metastasis early in their disease course. Therefore, incorporating PlGF expression as a prognostic tool could significantly improve the ability to predict metastatic potential and guide clinical decision-making for high-risk patients.
- Elaborate about the other diagnostic and theranostic tools used to diagnose and cure prostate cancer in introduction part.
Reply: Thanks for this point out. In the revised version of our manuscript the introduction has been improved as suggested.
The text was modified as follow:
Pages 1 and 2
Early-stage skeletal lesions are generally not detectable by current diagnostic tools. The sensitivity and specificity of these diagnostic tools are further limited when lesions progress slowly and may mimic non-malignant conditions. Consequently, the early de-tection of cancer cells’ molecular characteristics of localized disease could offer the identification of new theranostic tools useful in both prognosis and expanding thera-peutic options, thereby improving the life expectancy of patients with PCa while pre-venting the onset of metastatic disease [3-6].
- Kindly Change the funding details to “no funding” instead of “no founding”.
Reply: done
- Research work was perfectly explained in a reader friendly language.
- Images were of high quality and were brilliantly explained.
- Overall a great research paper.
Reply points #6, #7 and #8: Thank you again to the reviewer for their valuable comments.

Round 2
Reviewer 1 Report
Comments and Suggestions for Authors
Authors have addressed the comments